# Two-Time Correlation Functions in Dissipative and Interacting Bose–Hubbard Chains

**Zakari Denis** [1,2] **and Sandro Wimberger** [1,3,*] 

1   Dipartimento di Scienze Matematiche, Fisiche e Informatiche, Università di Parma, Campus Universitario, Parco Area delle Scienze n. 7/a, 43124 Parma, Italy; zakari.denis@u-psud.fr

2   International Centre for Fundamental Physics, Département de Physique, École Normale Supérieure, 24 rue Lhomond, 75005 Paris, France

3   Istituto Nazionale di Fisica Nucleare, Sezione di Milano Bicocca, Gruppo Collegato di Parma, 43124 Parma, Italy

*   Correspondence: sandromarcel.wimberger@unipr.it

**Abstract:** A method is presented for the systematic derivation of a hierarchy of coupled equations for the computation of two-time correlation functions of operators for open many-body quantum systems. We show how these systems of equations can be closed in mean-field and beyond approximations. Results for the specific example of the spectral weight functions are discussed. Our method allows one to access the full temporal evolution, not just the stationary solution, of non-equilibrium open quantum problems described by a Markovian master equation.

**Keywords:** open quantum systems; Bose–Einstein condensates; ultracold atoms; correlation functions; Green functions

## 1. Introduction

A very controlled way of introducing non-trivial dynamics to a Bose–Einstein condensate is to deplete a narrow region of the condensate by localised loss and watch the subsequent evolution of the many-body quantum system. For the experimental situation, as realised, e.g., in Herwig Otts's group at Kaiserslautern, atoms are ionised in a controlled way by an electronic beam and the produced ions and electrons are quickly extracted [1–5]. Consequently, there is scarcely any backaction onto the remaining atoms in the Bose condensate, provided that the filling factors (particle numbers per site) along the lattice are large. For such a setup, we can assume a Markovian coupling of the system to the environment. A corresponding Markovian master equation was used for such systems, taking into account localised loss in the lattice and phase noise, arising from interactions with the background gas or other experimental imperfections [6,7].

For small systems, typically two to four lattice sites, yet with reasonably large filling factors, we can unravel the Master equation exactly, using quantum jump Monte Carlo simulations [7–9]. For larger system sizes, approximative stochastic methods such as the truncated Wigner have been used to compute the single-particle density matrix (SPDM) [10,11] and normally ordered two-time correlation functions can be calculated in the Glauber–Sudarshan and positive-P representations [12,13]. Another beyond-mean-field method was successfully applied to propagate specifically chosen initial conditions, typically fully coherent Bose condensed states. This so-called Bogoliubov Back-reaction (BBR) method is based on a Bogoliubov–Born–Green–Kirkwood–Yvon (BBGKY) hierarchy expansion, consisting of dynamical equations for two-, four-, etc., point *equal-time* correlation functions, i.e., the expectation values of the two-, four-, etc., body reduced density matrices [6,14]. The interaction term in the Hubbard–Hamiltonian then induces the coupling between these dynamical equations toward higher orders. Typically, one truncates at the second order, approximating the six-point correlator by products

of two- and four-point correlation functions, in the way the moments of Gaussian variables would exactly split. This allows one to arrive at a closed system of coupled equations, which we can subsequently solve [7].

In this paper, we present a method to compute *non-equal time* correlation functions of operators, much in the spirit of the BBGKY hierarchy truncation for equal-time observables, which allows one to take into account higher orders in the fluctuations in a systematic way. To do so, we adapt the quantum regression theorem, extensively used in quantum optics, see e.g., [15–17], to interacting ultracold atoms in an open system's setting. Here, atom–atom interactions play an important role and result in hierarchies of dynamical equations that have to be truncated. More precisely, we want to compute two-time Green functions that are heavily used in solid-state physics, typically for fermonic transport problems, see e.g., [18–20]. In contrast to the latter applications, our open quantum systems are not time-translational invariant, which discards working in Fourier (frequency or energy) space and mapping the equations of motions into purely alegabric equations, as done e.g., in Ref. [21].

The paper is organised as follows: Section 2 introduces our open many-body boson system. Section 3 presents the computation scheme for two-time correlation functions, which is then applied in Section 4 in mean-field approximation. Section 5 discusses the next order beyond mean-field, with similar equations presented in Section 6 for the density–density correlation functions. The last Section 7 concludes the paper.

## 2. Dissipative Finite Bose–Hubbard Chain

We model ultracold bosons in sufficiently deep optical lattices by a tight-binding approximation, using a single-band Bose–Hubbard model. The geometry is assumed to be quasi one-dimensional, corresponding to a cigar-shaped confinement of the atoms (which is much stronger in the radial direction). Then, the dynamics of coherent interacting ultracold atoms tunnelling through an $M$-site lattice is described by the Bose–Hubbard Hamiltonian [22]:

$$\hat{H}_{\text{BH}} = -J \sum_{j=1}^{M-1} (\hat{a}_{j+1}^\dagger \hat{a}_j + \hat{a}_j^\dagger \hat{a}_{j+1}) + \frac{U}{2} \sum_{j=1}^{M} \hat{a}_j^\dagger \hat{a}_j^\dagger \hat{a}_j \hat{a}_j . \tag{1}$$

Here, $\hat{a}_j^\dagger$ and $\hat{a}_j$ denote respectively the creation and annihilation operators at site $j$, $J$ is the tunnelling rate and $U$ is the interaction strength. The reduced Planck's constant $\hbar$ is set to one, which corresponds to measuring all energies in frequency units. The natural unit of time is then $J^{-1}$.

Dissipation processes are accounted for by introducing the master equation in Lindblad form [17] together with a suitable Liouville superoperator:

$$\partial_t \hat{\rho}(t) = \mathcal{L}\hat{\rho}(t) = -i\big[\hat{H}_{\text{BH}}, \hat{\rho}(t)\big] + \widetilde{\mathcal{L}}\hat{\rho}(t), \tag{2}$$

$$\widetilde{\mathcal{L}}\hat{\rho} = -\sum_{j=1}^{M} \frac{\gamma_j}{2} \big(\hat{L}_j^\dagger \hat{L}_j \hat{\rho} + \hat{\rho}\hat{L}_j^\dagger \hat{L}_j - 2\hat{L}_j \hat{\rho}\hat{L}_j^\dagger\big). \tag{3}$$

The Lindblad operators $\hat{L}_j$ are chosen depending on the type of relaxation or decoherence process relevant for the system under study. In the following, we restrict to local single-body dissipation as motivated in the introduction, i.e., $\hat{L}_j = \hat{a}_j$, and $\gamma_j$ is the dissipation rate at site $j$. This choice of the Lindblad operator, as shown in [7], leads to equations of motion for the SPDM equivalent to the heuristic non-Hermitian discrete nonlinear Schrödinger equation introduced in [23] and successfully applied to the description of localized single-body dissipation processes in Bose–Hubbard chains in good agreement with experimental realizations [3,5]. Further sets of Lindblad operators modelling other processes in Bose–Hubbard chains can be found in the review [24].

The expectation value of some system operator $\hat{A}$ is then provided by the trace $\langle \hat{A}(t)\rangle = \text{Tr}\{\hat{A}\hat{\rho}(t)\} = \text{Tr}\{\hat{A}_H(t)\hat{\rho}(0)\}$. From this, the equivalent Heisenberg representation is defined as $\hat{A}_H(t) = V^\dagger(t,0)\hat{A}$, with the propagator $V(t,t_0) = \exp((t-t_0)\mathcal{L})$, for a time-independent

Liouvillian. Henceforth, the Heisenberg representation is not specified by an index but simply indicated by the presence of a time argument. In this representation, the time evolution of the Heisenberg operator $\hat{A}(t)$ is carried out by the adjoint master equation:

$$\partial_t \hat{A}(t) = V^\dagger(t,0)(\mathcal{L}^\dagger \hat{A}) = i[\hat{H}_{\text{BH}}, \hat{A}](t) + (\widetilde{\mathcal{L}}^\dagger \hat{A})(t), \tag{4}$$

$$\widetilde{\mathcal{L}}^\dagger \hat{A} = -\sum_{j=1}^{M} \frac{\gamma_j}{2} \left(\hat{L}_j^\dagger \hat{L}_j \hat{A} + \hat{A}\hat{L}_j^\dagger \hat{L}_j - 2\hat{L}_j^\dagger \hat{A}\hat{L}_j\right) = -\sum_{j=1}^{M} \frac{\gamma_j}{2} \left(\hat{L}_j^\dagger [\hat{L}_j, \hat{A}] + [\hat{A}, \hat{L}_j^\dagger]\hat{L}_j\right). \tag{5}$$

The main observable in the introduced system is the single-particle density matrix (SPDM) $\sigma_{j,k} = \langle \hat{a}_j^\dagger \hat{a}_k \rangle = \text{Tr}\{\hat{a}_j^\dagger \hat{a}_k \hat{\rho}\}$. Its diagonal matrix elements give the local populations of the chain, while its off-diagonal elements provide information about the coherence of the state [7]. The quartic interaction term leads to a coupling of the equations of motion verified by the SPDM and higher-order moments. Consequently, computing the SPDM requires a prior truncation of the hierarchy into a closed set of differential equations. The mean-field (MF) approximation keeps only the first order of the hierarchy by neglecting the covariances $\Delta_{jmkn} = \langle \hat{a}_j^\dagger \hat{a}_m \hat{a}_k^\dagger \hat{a}_n \rangle - \langle \hat{a}_j^\dagger \hat{a}_m \rangle \langle \hat{a}_k^\dagger \hat{a}_n \rangle$, whereas the BBR close to mean-field method evolves simultaneously the SPDM and the covariance, truncating higher-order moments [6,7,14].

The aim of this article is to provide a method for obtaining multi-time correlation functions for quantum many-body systems from the knowledge of the equal-time correlation functions. Our method works for systems with relaxation channels according to the master Equation (3) and gives access to the transient dynamics of the functions, not just to the stationary-state solutions.

## 3. Computation Scheme for Two-Point Correlation Functions

Two-point Green functions (GF) provide useful information about many-body systems such as temporal and spatial correlations or information about the response of the system to an external perturbation [20]. As a first example, if one is interested in particular in evaluating the average probability of a particle propagating from some site $k$ at time $t$ to any other site $j$ at time $t' = t + \tau$, the retarded Green function provides relevant information. In this case, this correlation function reads:

$$G_{j,k}^R(t + \tau; t) = \theta_{\text{H}}(\tau)(G_{j,k}^>(t + \tau; t) - G_{j,k}^<(t + \tau; t)), \tag{6}$$

where the *lesser* and *greater* bosonic GFs are defined as follows:

$$G_{j,k}^<(t'; t) = -i\langle \hat{a}_k^\dagger(t)\hat{a}_j(t')\rangle, \qquad G_{j,k}^>(t'; t) = -i\langle \hat{a}_j(t')\hat{a}_k^\dagger(t)\rangle, \tag{7}$$

where the ladder operators are expressed in the above-recalled Heisenberg representation $V^\dagger(t,0)\hat{a}_j^{(\dagger)}$. In order to compute these last two GFs, we derive a closed set of equations of motion for a dissipative setting by means of a modified version of the quantum regression theorem.

*Quantum Regression Hierarchy*

Let $\{\hat{A}_i\}$ be a set of arbitrary system operators, e.g., $\hat{n}_i = \hat{a}_i^\dagger \hat{a}_i$, and $\mathcal{L}$ a time-independent Liouville superoperator, e.g., the one defined in Equations (4) and (5), the adjoint master equation reads:

$$\partial_t \hat{A}_i(t) = V^\dagger(t,0)(\mathcal{L}^\dagger \hat{A}_i). \tag{8}$$

Let us now make the general assumption that the adjoint Liouville operator acts on $\hat{A}_i$ in such a way that its expectation value can be rewritten as:

$$\mathrm{Tr}\{(\mathcal{L}^\dagger \hat{A}_i)\hat{\rho}(t)\} = \mathrm{Tr}\left\{\left(\sum_\ell T_{i\ell}^{(1)}\hat{A}_\ell + \sum_{\ell,\ell'} K_{i\ell'\ell}^{(1)}\hat{B}_{\ell'}\hat{A}_\ell\right)\hat{\rho}(t)\right\}, \tag{9}$$

where $\{\hat{B}'_\ell\}$ are operators composed of an *even* number of creation and annihilation operators, e.g., a power of the density operator. This is the typical relation one gets when dealing with non-quadratic Hamiltonians and/or nonlinear Lindbald operators, and in particular the previously defined Liouvillian, which result in a coupling between the equations of motion satisfied by $n$-point and higher-order correlation functions. This relation is assumed to hold for any initial density matrix $\hat{\rho}$ so that one is able to identify both operators in parentheses:

$$\mathcal{L}^\dagger \hat{A}_i = \sum_\ell T_{i\ell}^{(1)}\hat{A}_\ell + \sum_{\ell,\ell'} K_{i\ell'\ell}^{(1)}\hat{B}_{\ell'}\hat{A}_\ell. \tag{10}$$

The *K*-term requires an extension of the quantum regression theorem (see [25,26] or the Section 5.2.3 in [16] and Section 3.2.3 in [17]). Combining Equations (8) and (10), the equation of motion of the two-point correlation function reads:

$$\begin{aligned}
\partial_\tau \langle \hat{A}_i(t+\tau)\hat{A}_j(t)\rangle &= \mathrm{Tr}\{(\mathcal{L}^\dagger \hat{A}_i)V(t+\tau,t)\hat{A}_j V(t,0)\hat{\rho}(0)\} = \langle (\mathcal{L}^\dagger \hat{A}_i)V(t+\tau,t)\hat{A}_j V(t,0)\rangle \\
&= \sum_\ell T_{i\ell}^{(1)}\langle \hat{A}_\ell(t+\tau)\hat{A}_j(t)\rangle + \sum_{\ell,\ell'} K_{i\ell'\ell}^{(1)}\langle (\hat{B}_{\ell'}\hat{A}_\ell)(t+\tau)\hat{A}_j(t)\rangle \\
&= \sum_\ell T_{i\ell}'^{(1)}(t+\tau)\langle \hat{A}_\ell(t+\tau)\hat{A}_j(t)\rangle + \sum_{\ell,\ell'} K_{i\ell'\ell}^{(1)}\langle (\Delta\hat{B}_{\ell'}\hat{A}_\ell)(t+\tau)\hat{A}_j(t)\rangle,
\end{aligned} \tag{11}$$

where $T_{i\ell}'^{(1)}(t) = T_{i\ell}^{(1)} + \sum_{\ell'} K_{i\ell'\ell}^{(1)}\langle \hat{B}_{\ell'}(t)\rangle$ and the central moment operator is defined as $\Delta\hat{A} = \hat{A} - \langle\hat{A}\rangle$. Naturally, the same can be done for $\langle \hat{A}_j(t)\hat{A}_i(t+\tau)\rangle$, with the only difference being that $\hat{A}_j$ is then placed at the left end of the moments.

With this proper rewriting, one gets a hierarchy of coupled dynamical equations in the form of the BBGKY hierarchy:

$$\partial_\tau \langle \hat{A}_i(t+\tau)\hat{A}_j(t)\rangle = \sum_\ell T_{i\ell}'^{(1)}(t+\tau)\langle \hat{A}_\ell(t+\tau)\hat{A}_j(t)\rangle + \sum_{\ell,\ell'} K_{i\ell'\ell}^{(1)}\langle (\Delta\hat{B}_{\ell'}\hat{A}_\ell)(t+\tau)\hat{A}_j(t)\rangle, \tag{12}$$

$$\begin{aligned}
\partial_\tau \langle (\Delta\hat{B}_{i'}\hat{A}_i)(t+\tau)\hat{A}_j(t)\rangle &= \sum_\ell T_{i\ell}'^{(2)}(t+\tau)\langle (\Delta\hat{A}_{\ell'}\hat{A}_\ell)(t+\tau)\hat{A}_j(t)\rangle \\
&\quad + \sum_{\ell,\ell',k} K_{i'i\ell'\ell k}^{(2)}\langle (\Delta\hat{B}_k\Delta\hat{B}_{\ell'}\hat{A}_\ell)(t+\tau)\hat{A}_j(t)\rangle.
\end{aligned} \tag{13}$$

$$\vdots$$

This system of differential equations can then be closed by truncating the hierarchy to some order in the fluctuations provided that the time-local averages $\langle \hat{B}_{\ell'}(t)\rangle$ can be calculated at any time $t$. In the simplest case, for which $K^{(1)}$ is a zero matrix, one naturally recovers the standard quantum regression theorem expression. Otherwise, the first approximation is to make a mean-field approximation and neglect the covariances of the operators $\hat{B}_{\ell'}$ and $(\hat{A}_\ell\hat{A}_j)$ in Equation (12). To perform this approximation, it is convenient to first rewrite Equation (12) in such a way that the order in the fluctuations is explicit:

$$\partial_\tau \langle \hat{A}_i(t+\tau)\hat{A}_j(t)\rangle = \sum_\ell T_{i\ell}'^{(1)}(t+\tau)\langle \hat{A}_\ell(t+\tau)\hat{A}_j(t)\rangle \tag{14}$$

$$+ \sum_{\ell,\ell'} K_{i\ell'\ell}^{(1)}\langle \Delta\hat{B}_{\ell'}(t+\tau)\Delta(\hat{A}_\ell(t+\tau)\hat{A}_j(t))\rangle. \tag{15}$$

From this expression, one observes that here the mean-field approximation amounts to keeping only the line (14), which results in a quantum regression expression with a time-dependent

coefficient matrix $T'^{(1)}$. The knowledge of $T'^{(1)}$ at any time puts then the system into a closed form. Moreover, if one is able to compute $\langle \hat{A}_{\ell'} \rangle$ and $\langle \hat{A}_\ell \hat{A}_j \rangle$ at any time at second order in the fluctuations, for instance by employing the BBR truncation [7], then one can obtain an approximation of the GFs at this order by including the line (15) and computing the Equation (13) of the hierarchy in the BBR approximation.

## 4. Mean-Field Approximation

We study now the Bose–Hubbard Hamiltonian of Equation (1) with single-body loss putting $\hat{L}_j = \hat{a}_j$ in Equation (2). In the mean-field approximation, the equations of motion of the annihilation operators read in that case

$$\partial_t \hat{a}_j(t) = iJ(\hat{a}_{j+1}(t) + \hat{a}_{j-1}(t)) - iU\hat{a}_j^\dagger(t)\hat{a}_j(t)\hat{a}_j(t) - \frac{\gamma_j}{2}\hat{a}_j(t)$$

$$= \sum_\ell \underbrace{\left(iJ(\delta_{j+1,\ell} + \delta_{j-1,\ell}) - \frac{\gamma_\ell}{2}\delta_{j,\ell}\right)}_{T_{j\ell}} \hat{a}_\ell(t) + \sum_{\ell,\ell'} \underbrace{\left(-iU\delta_{j,\ell'}\delta_{\ell',\ell}\right)}_{K_{j\ell'\ell}} \hat{n}_{\ell'}(t)\hat{a}_\ell(t) \qquad (16)$$

$$= \sum_\ell T'_{j\ell}(t)\hat{a}_\ell(t) + \sum_{\ell,\ell'} K_{j\ell'\ell}\Delta\hat{n}_{\ell'}(t)\hat{a}_\ell(t),$$

where

$$T'_{j\ell}(t) = iJ(\delta_{j+1,\ell} + \delta_{j-1,\ell}) - iU\delta_{j,\ell}n_\ell(t) - \frac{\gamma_\ell}{2}\delta_{j,\ell}. \qquad (17)$$

The simplicity of the equation of motion satisfied by the annihilation operator allows one to get directly a relation of the form of Equation (10), without the need of considering its expectation value and taking advantage of the cyclic permutation invariance of the trace.

Then, the quantum regression theorem yields:

$$\partial_t \hat{a}_j(t) \stackrel{\text{MF}}{\approx} \sum_\ell T'_{j\ell}(t)\hat{a}_\ell(t) \quad \Rightarrow \quad \begin{cases} \partial_\tau \langle \hat{a}_k^\dagger(t)\hat{a}_j(t+\tau)\rangle \stackrel{\text{MF}}{\approx} \sum_\ell T'_{j\ell}(t+\tau)\langle \hat{a}_k^\dagger(t)\hat{a}_\ell(t+\tau)\rangle, \\ \partial_\tau \langle \hat{a}_j(t+\tau)\hat{a}_k^\dagger(t)\rangle \stackrel{\text{MF}}{\approx} \sum_\ell T'_{j\ell}(t+\tau)\langle \hat{a}_\ell(t+\tau)\hat{a}_k^\dagger(t)\rangle. \end{cases} \qquad (18)$$

Due to the dependence of $T'$ on the local density $n_\ell = \langle \hat{a}_\ell^\dagger \hat{a}_\ell \rangle = \sigma_{\ell\ell}$ these equations of motion have to be evolved along with those of the single particle density matrix. This leads to the following closed set of differential equations:

$$i\partial_t \sigma_{j,k} \stackrel{\text{MF}}{\approx} -J(\sigma_{j,k+1} + \sigma_{j,k-1} - \sigma_{j+1,k} - \sigma_{j-1,k}) + U(n_j - n_k)\sigma_{j,k} - \frac{i}{2}(\gamma_j + \gamma_k)\sigma_{j,k}, \qquad (19)$$

$$i\partial_\tau G_{j,k}^{\lessgtr}(t+\tau;t) \stackrel{\text{MF}}{\approx} -J\big(G_{j+1,k}^{\lessgtr}(t+\tau;t) + G_{j-1,k}^{\lessgtr}(t+\tau;t)\big) + \big(Un_j(t+\tau) - i\frac{\gamma_j}{2}\big)G_{j,k}^{\lessgtr}(t+\tau;t). \qquad (20)$$

These can be evolved in $\tau \geq 0$ for each $t$. In this way, the first time argument is always later than the second one, but this is not restrictive as the opposite situation can be obtained from the relation:

$$G_{j,k}^{\lessgtr}(t;t+\tau) = -(G_{k,j}^{\lessgtr}(t+\tau;t))^*. \qquad (21)$$

*Illustrative Results*

As an example of the performance of the method presented here, we compute the correlation function defined as $A_{j,k}(t',t) = \langle [\hat{a}_j(t'), \hat{a}_k^\dagger(t)]\rangle = i(G_{j,k}^>(t';t) - G_{j,k}^<(t';t)) = i(G_{j,k}^R(t';t) - G_{j,k}^A(t';t))$ for two different settings. The Fourier transform of $A_{j,k}(t',t)$ presents the spectral weight function [20], used e.g., for computing currents in transport setups. This particular correlation function is chosen because its equal-time values and its lower and upper bounds can readily be checked and because its magnitude is equal to that of the retarded Green's function for $t' > t$, whereas the magnitude of the advanced Green's function is given by the $t > t'$ half-quadrant. In addition, in the noninteracting

non-dissipative case, its profile simply consists of periodic oscillations extending to both sides away from the diagonal $t = t'$. This harmonic behaviour of $A_{j,k}(t', t)$ is related to the mentioned fact that its Fourier transform characterises the spectrum without perturbation in the closed system's case, in the example shown below of a three-level system.

Figure 1 shows the magnitude of two matrix elements of the spectral weight function of a condensate initially made of $N_0 = 1000$ atoms loaded into a three-well Bose–Hubbard chain with an initial population imbalance between a low populated central well 2 and its two more populated neighbours 1 and 3 in the presence of strong dissipation occurring at the central site. The values taken at the diagonal $t = t'$ correspond to the constant $|A_{j,k}| = |\langle[\hat{a}_j(t), \hat{a}_k^\dagger(t)]\rangle| = \delta_{j,k}$, as expected. In Figure 1a, which depicts the $|A_{2,1}|$ matrix element, the $t' > t$ half-quadrant presents an elevation at early $t'$ that drops as $t'$ becomes larger than a few $J^{-1}$. This indicates that, on average, the flow of particles from the site 1 to the leaky site 2 is suppressed at times above this typical value.

This suppression is a signature of the so-called quantum Zeno effect, meaning that particles are blocked on average from flowing into the site with dissipation, see e.g., the results and descriptions in [3,24,27]. Dissipation can thus induce an effectively self-trapped regime, considerably lowering the average inter-well tunnelling, although the values of the population imbalance and the interaction strength do not suffice to reach this regime in the absence of dissipation. In Figure 1b, which depicts the $|A_{2,1}|$ matrix element, whereas, in the noninteracting non-dissipative case, this function shows periodic oscillations from either side of the diagonal around the value $1/2$; in this case, the spectral weight function quickly attains a plateau at this value. The fact that it does not take values below one half after a few $J^{-1}$ is another signature of the quenching of the inter-well tunnelling induced by a quantum Zeno effect.

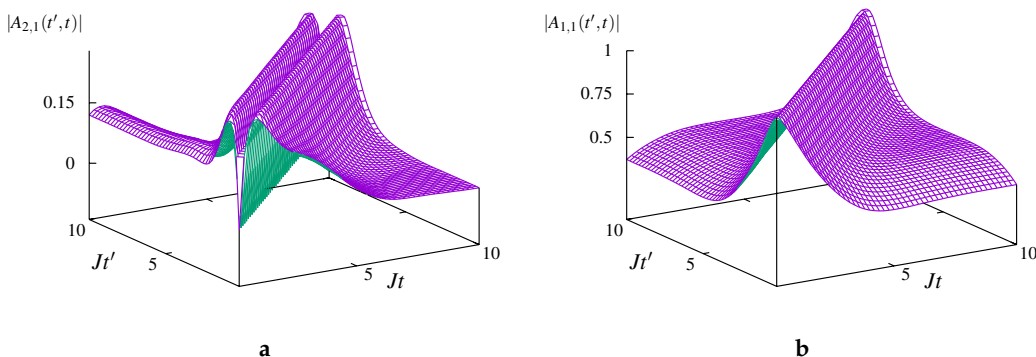

**Figure 1.** Magnitude of the spectral weight function $A_{j,k}$ for an initially pure Bose–Einstein condensate with $n_1(0) = 500$, $n_2(0) = 50$, $n_3(0) = 450$, and interaction $UN_0/J = 10$; the dissipation rate is set to $\gamma_2/J = 5$. (**a**) interwell correlations $|A_{2,1}(t', t)|$; (**b**) onsite correlations $|A_{1,1}(t', t)|$.

Figure 2 represents the same correlation functions for the same three-well setting but in the absence of dissipation. In this case, the profile of the spectral weight function is very rugged due to the interactions. The correlation function at $t \neq t'$ fluctuates around $1/2$ regardless of the matrix element, which indicates that the system is *not* in a self-trapped regime, as the average inter-site flow of bosons is not suppressed.

From these two examples, one observes that correlations in Bose–Hubbard chains qualitatively differ depending on the presence or absence of dissipation. This as well as the observation of some particular structures in the profiles of the investigated correlation functions can be used to determine and differentiate the dynamical regimes in Bose–Hubbard chain set-ups, where the interplay of interactions and dissipation plays a major role.

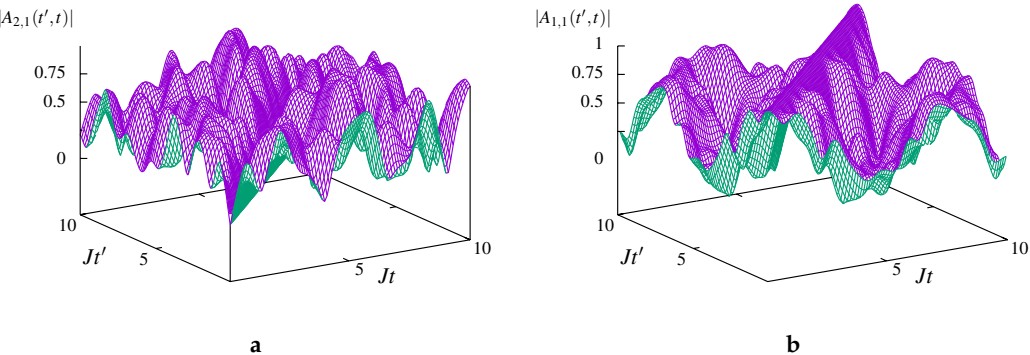

**Figure 2.** Magnitude of the spectral weight function $A_{j,k}$ for an initially pure Bose–Einstein condensate with $n_1(0) = 500$, $n_2(0) = 50$, $n_3(0) = 450$, and interaction $UN_0/J = 10$; the dissipation rate is set to zero, $\gamma_2 = 0$. (**a**) interwell correlations $|A_{2,1}(t',t)|$; (**b**) onsite correlations $|A_{1,1}(t',t)|$.

## 5. Beyond Mean-Field Approximation

One of the assets that motivate seeking correlation functions as solutions to a hierarchy of coupled equations of motion is that the contribution of higher order fluctuations can be integrated in a systematic way—for instance, by taking into account the second equation, Equation (13), of the hierarchy and truncating sextic moments as follows [6,14]:

$$\langle \hat{a}_j^\dagger \hat{a}_m \hat{a}_k^\dagger \hat{a}_n \hat{a}_r^\dagger \hat{a}_s \rangle \overset{\text{BBR}}{\approx} \langle \hat{a}_j^\dagger \hat{a}_m \hat{a}_k^\dagger \hat{a}_n \rangle \langle \hat{a}_r^\dagger \hat{a}_s \rangle + \langle \hat{a}_j^\dagger \hat{a}_m \hat{a}_r^\dagger \hat{a}_s \rangle \langle \hat{a}_k^\dagger \hat{a}_n \rangle + \langle \hat{a}_k^\dagger \hat{a}_n \hat{a}_r^\dagger \hat{a}_s \rangle \langle \hat{a}_j^\dagger \hat{a}_m \rangle - 2\langle \hat{a}_j^\dagger \hat{a}_m \rangle \langle \hat{a}_k^\dagger \hat{a}_n \rangle \langle \hat{a}_r^\dagger \hat{a}_s \rangle. \tag{22}$$

One gets a closed set of differential equations for the GFs in the Bogoliubov Back-reaction beyond mean-field approximation. For example, for the greater GF, this reads:

$$i\partial_\tau G_{j,k}^{>}(t+\tau;t) \overset{\text{BBR}}{\approx} -J\big(G_{j+1,k}^{>}(t+\tau;t) + G_{j-1,k}^{>}(t+\tau;t)\big) + \big(Un_j(t+\tau) - i\tfrac{\gamma_j}{2}\big)G_{j,k}^{>}(t+\tau;t) \\ + U F_{j,j,j;k}^{>}(t+\tau;t), \tag{23}$$

$$i\partial_\tau F_{j,m,n;k}^{>}(t+\tau;t) \overset{\text{BBR}}{\approx} + J\big( F_{j+1,m,n;k}^{>}(t+\tau;t) + F_{j-1,m,n;k}^{>}(t+\tau;t) - F_{j,m+1,n;k}^{>}(t+\tau;t) \\ - F_{j,m-1,n;k}^{>}(t+\tau;t) - F_{j,m,n+1;k}^{>}(t+\tau;t) - F_{j,m,n-1;k}^{>}(t+\tau;t)\big) \\ - U\big( F_{j,j,n;k}^{>}(t+\tau;t)\sigma_{jm}(t+\tau) - (\Delta_{j,m,m,m} + \Delta_{j,m,n,n})(t+\tau)iG_{n,k}^{>}(t+\tau;t) \\ + F_{j,m,n;k}^{>}(t+\tau;t)(n_j - n_m - n_n)(t+\tau) \\ + \delta_{m,n}(F_{j,m,n;k}^{>}(t+\tau;t) + \sigma_{j,m}G_{n,k}^{>}(t+\tau;t))\big). \tag{24}$$

Here, we defined

$$iF_{j,m,n;k}^{>}(t';t) = \langle \Delta(\hat{a}_j^\dagger \hat{a}_m)(t')\Delta(\hat{a}_n(t')\hat{a}_k^\dagger(t)) \rangle = \langle (\hat{a}_j^\dagger \hat{a}_m \hat{a}_n)(t')\hat{a}_k^\dagger(t) \rangle - \sigma_{j,m}(t')iG_{n,k}^{>}(t';t). \tag{25}$$

The SPDM and the covariances are given in the BBR approximation by (c.f. [7]):

$$i\partial_t \sigma_{jk} \overset{\text{BBR}}{\approx} -J\big(\sigma_{j,k+1} + \sigma_{j,k-1} - \sigma_{j+1,k} - \sigma_{j-1,k}\big) + U\big(\Delta_{jkkk} + \sigma_{jk}\sigma_{kk} - \Delta_{jjjk} - \sigma_{jj}\sigma_{jk}\big) - i\frac{\gamma_j + \gamma_k}{2}\sigma_{j,k}, \tag{26}$$

$$i\partial_t \Delta_{jmkn} \overset{\text{BBR}}{\approx}$$
$$-J\big(\Delta_{j,m,k,n+1} + \Delta_{j,m,k,n-1} + \Delta_{j,m+1,k,n} + \Delta_{j,m-1,k,n} - \Delta_{j,m,k+1,n} - \Delta_{j,m,k-1,n} - \Delta_{j+1,m,k,n} - \Delta_{j-1,m,k,n}\big)$$
$$+ U\big(\sigma_{jm}(\Delta_{mmkn} - \Delta_{jjkn}) + \sigma_{kn}(\Delta_{jmnn} - \Delta_{jmkk}) + \Delta_{jmkn}(-\sigma_{jj} + \sigma_{mm} - \sigma_{kk} + \sigma_{nn})\big)$$
$$- i\frac{\gamma_j + \gamma_m + \gamma_k + \gamma_n}{2}\Delta_{jmkn}. \tag{27}$$

However, although being more accurate in the low population regime, no qualitative difference is expected with respect to the mean-field version, at least in the limit of large filling factors $n_j \gg 1$.

## 6. Density–Density Correlation Function

Besides improving the mean-field precision, BBR also provides a way to compute non-trivial density–density correlation functions. The latter indicate, for instance, possible temporal bunching or anti-bunching effects [7,15]. Indeed, at the mean-field level, one just gets $\langle (\hat{a}_j^\dagger \hat{a}_m)(t')(\hat{a}_k^\dagger \hat{a}_n)(t)\rangle \overset{\text{MF}}{\approx} \sigma_{jm}(t')\sigma_{kn}(t)$. Instead, by defining the density–density correlation function as follows:

$$C_{j,m;k,n}(t';t) = -i\langle \Delta(\hat{a}_j^\dagger \hat{a}_m)(t')\Delta(\hat{a}_k^\dagger \hat{a}_n)(t)\rangle = \langle (\hat{a}_j^\dagger \hat{a}_m)(t')(\hat{a}_k^\dagger \hat{a}_n)(t)\rangle - \sigma_{jm}(t')\sigma_{kn}(t) \qquad (28)$$

and truncating sextic moments according to Equation (22) in the equation of motion similar to Equation (12) that it satisfies, one gets:

$$
\begin{aligned}
i\partial_\tau C_{j,m;k,n}(t+\tau;t) \overset{\text{BBR}}{\approx} \\
&+ J\big(C_{j+1,m;k,n}(t+\tau;t) + C_{j-1,m;k,n}(t+\tau;t) - C_{j,m+1;k,n}(t+\tau;t) - C_{j,m-1;k,n}(t+\tau;t)\big) \\
&- U\big((n_j - n_m)(t+\tau)C_{j,m;k,n}(t+\tau;t) + \sigma_{jm}(C_{j,j;k,n}(t+\tau;t) - C_{m,m;k,n}(t+\tau;t))\big) \\
&- i\frac{\gamma_j + \gamma_m}{2}C_{j,m;k,n}(t+\tau;t),
\end{aligned}
\qquad (29)
$$

which has to be evolved together with Equations (26) and (27). Again, if one wants to compute the density–density correlation function with the first time argument evaluated at an earlier time than the second, one can simply use

$$C_{j,m;k,n}(t;t+\tau) = -\big(C_{n,k;m,j}(t;t+\tau)\big)^*. \qquad (30)$$

This shows that the systematic expansions in order to approximate the temporal correlation functions of operators proposed here are indeed very useful to compute physical relevant quantities in open quantum many-body systems.

## 7. Conclusions

Usually for a quantum many-body problem, the Schrödinger equation, or in our case the master equation Equation (2), is not analytically solvable. One possible way to access the non-equilibrium dynamics of such systems is with the help of Green functions. The two-time correlators, such as the time-dependent first order coherences from Equations (6) and (7), or the density–density correlation functions Equation (28), allow one to characterize the out-of-equilibrium dynamics of many-body interacting quantum systems. We have presented a method on how to systematically compute a hierarchy of approximations for two-time correlation functions, in principle of arbitrary operators, which applies to a wide class of interacting systems whose dissipation is described by a master equation of the form of Equation (2). Based on our method, we can treat particle interactions on a mean-field level (see Section 4) and also one-order beyond (see Section 5). Higher-order expansions are possible, at the cost of very lengthy formulae, which at some point will become difficult to deal with even numerically (see e.g., [28,29] for just time-equal correlators).

With our method, we computed the spectral weight function in Section 4 as an example of a physical observable. In principle, we can compute the temporal evolution and the correlations of many interesting quantities for experiments, such as the onsite populations or the density–density correlations between two lattice sites. For about ten years, state-of-the-art experiments with ultracold atoms can access information on static correlation functions with high spatial resolution (see e.g., [30–36]). Time-dependent two-point correlation functions can be measured too, see e.g., [37], which would allow for direct applications of our developed theoretical approach.

Our problem is defined by a time-local master equation, which means that the coupling to the environment, in our case a zero-temperature sink, is supposed to be sufficiently weak for justifying the Markovian assumption underlying Equation (2). Strong coupling to the environment, such as that present in lead-to-lead transport across solid-state samples (see e.g., [18,19]) remains an open problem since then we may not apply the quantum regression theorem valid for time-local master equations. For tedious extensions of the theorem to non Markovian setups, the reader may consult e.g., [38–41]. An alternative approach would be then to fully include the leads into the treatment and use a diagrammatic expansion of the Green functions computed for the full lead-system-lead system, such as done for bosons in steady state in Refs. [42,43]. Needless to say, such an approach as just mentioned is very hard for many-body quantum systems whose main central part, without the leads, is itself non-integrable, such as our many-body Bose–Hubbard model for more than two wells [44,45].

**Acknowledgments:** We thank Andreas Komnik for initiating this project. Sandro Wimberger is very grateful to the organisers of the conference SuperFluctuations 2017 for his invitation.

**Author Contributions:** The authors contributed equally to this work.

**Conflicts of Interest:** The authors declare no conflict of interest.

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
