# Peer review of "Two-Time Correlation Functions in Dissipative and Interacting Bose–Hubbard Chains"

_condensedmatter, doi:10.3390/condmat3010002_

Reviewer 1 Report

I think this is a good paper. It extends the BBR method to the approximate calculation of two-time correlation functions in non-stationary states, which are important in many physical studies. I found it interesting that the quantum regression theorem has been adapted for systems with nonlinear equations of motion. To illustrate their method, the authors have chosen what is now a fairly standard Bose-Hubbard trimer with dissipation. This is a worthwhile example. 

One minor point is that Gaussian should be capitalised.
I found little technically wrong with this paper apart from the statement immediately after Eq. 17 that the creation operator has zero expectation value. This is entirely state dependent and may be true in the long time limit of a system with defusing phase, but is certainly not true of coherent states.

There are, of course, other methods of calculating these two-time correlation functions, using stochastic integration in either the approximate truncated Wigner or the exact positive-P representations. For completeness, the authors might like to mention some of these, although this is completely up to them. 

The single-time SPDM has been calculated for similar systems in PRA 84 043636 (2011) and Eur. Phys. J. D 71 18 (2017). While it is possible to calculate a two-time SPDM in the truncated Wigner, it does become difficult. Interestingly enough, one of the quantities necessary to do this is exactly the two-time commutator of Heisenberg operators that the authors use, see PRA 80 033624 (2009).

Normally-ordered two-time correlation functions are calculated naturally in the Glauber-Sudarshan and positive-P representations, although the solving of these can present difficulties. PRA 95 0637 (2017) used this ability to calculate output spectra from damped BH wells. The averaged solutions from stochastic integration were taken to enable the system to be represented as an Ornstein-Uhlenbeck process, from which the spectra are calculated algebraically. 

I would like to wish the editors and authors Merry Christmas and Buon Natale.

Author Response

We thank the Referees for their very positive evaluation of our work. All changes are marked in red in the new manuscript.

Reply to Reviewer 1:

"Gaussian" has been capitalized now. The statement about the average value of the annihilation operator has been clarified into the new text after Eq. (17). All mentioned references are cited in the Introduction as alternative (approximative) methods for treating longer chains, see our second paragraph there.

Reviewer 2 Report

This paper has derived a two-point correlation function of annihilation and creation operators for the Bose-Hubbart model with linear dissipation in the mean field approximation. In this work, the Markovian time evolution is assumed for the system and so the dissipation is described by the phenomenological quantum master equation of the Lindblad form. First using the master equation, the authors have derived the equation of motion for the two-point correlation function of annihilation and creation operator, which is not a closed form. To obtain the closed set of equations, they have applied the mean field approximation. Then the resulting equations of motion are solved numerically. The authors have discussed the physical meaning of the results. Furthermore they have briefly considered the time evolution of the two-point correlation function beyond the mean field approximation.  I think that the results obtained in the present manuscript are sound and interesting. Therefore I would recommend the publication in Condensed Matter. Before publication, however, it would be better for the authors to explain the reason why the Lindblad operator is given by $L_{j}=a_{j}$. In general, considering an interaction system (the Bose-Hubbart model in this manuscript), one needs to find eigenvalues and eigenvectors of the system Hamiltonian to find the Lindblad operator (see Chapter 3 in Ref.[13]).

Author Response

We thank the Referees for their very positive evaluation of our work. All changes are marked in red in the new manuscript.

Reply to Reviewer 2:

The exact form of the Lindblad operator in the dissipative part of the master equation is irrelevant for the purpose of our new method which access two-time correlators. Nevertheless, we chose a specific example in order to simplify the presentation and the form of equations. Our choice is motivated by a series of previous papers and experiments using exactly such an heuristic single-particle decay process. Formally, we agree with the Referee that the Kraus operators appearing in the master equation are typically evaluated as matrix elements in the eigenbasis of the environment and the system up to some unitary transformation, and therefore, as soon as the state of the system deviates from a coherent state, it will not be an eigenstate of the annihilation operators chosen as Lindblad operator in this work. Single-body decay obviously is well defined only in the non-interacting limit of the Bose-Hubbard model, as its name already implies. Nevertheless, it realistically models a decay in this latter and also in the mean-field limit (within the so-called non-hermitian Gross-Pitaevskii equation). Since here we also work on extensions of the mean-field limit we should be on the safe side. More information on this type of single-particle decay is found in our references, in particular in the further papers cited in our ref. [7], in the experimental papers [1-6, 23] and the review [24], in the new numbering. We refer to them in the first paragraph of the Introduction and/or in the now extended paragraph after Eq. (3).